# Towards trustworthy predictions from deep neural networks with fast adversarial calibration

## Abstract

To facilitate a wide-spread acceptance of AI systems guiding decision making in real-world applications, trustworthiness of deployed models is key. That is, it is crucial for predictive models to be uncertainty-aware and yield well-calibrated (and thus trustworthy) predictions for both in-domain samples as well as under domain shift. Recent efforts to account for predictive uncertainty include post-processing steps for trained neural networks, Bayesian neural networks as well as alternative non-Bayesian approaches such as ensemble approaches and evidential deep learning. Here, we propose an efficient yet general modelling approach for obtaining well-calibrated, trustworthy probabilities for samples obtained after a domain shift. We introduce a new training strategy combining an entropy-encouraging loss term with an adversarial calibration loss term and demonstrate that this results in well-calibrated and technically trustworthy predictions for a wide range of perturbations. We comprehensively evaluate previously proposed approaches on different data modalities, a large range of data sets including sequence data, network architectures and perturbation strategies and observe that our modelling approach substantially outperforms existing state-of-the-art approaches, yielding well-calibrated predictions under domain drift.

## 1 Introduction

To facilitate a wide-spread acceptance of AI systems guiding decision making in real-world applications, trustworthiness of deployed models is key. Not only in safety-critical applications such as autonomous driving or medicine (Helldin et al., 2013; Caruana et al., 2015; Leibig et al., 2017), but also in dynamic open world systems in industry it is crucial for predictive models to be uncertainty-aware. Only if predictions are calibrated in the case of any gradual domain shift, covering the entire spectrum from in-domain ("known unknowns") to truly out-of-domain samples ("unknown unknowns"), they can be trusted. In particular in industrial and IoT settings, deployed models may encounter erroneous and inconsistent inputs far away from the input domain throughout the life-cycle; in addition, the distribution of the input data may gradually move away from the distribution of the training data (e.g. due to wear and tear of the assets, maintenance procedures or change in usage patterns).

A variety of approaches to account for predictive uncertainty exist. They include post-processing steps for trained neural networks, where for example a validation set, drawn from the same distribution as the training data, is used to rescale the logit vectors returned by a trained neural network such that in-domain predictions are well calibrated (Platt, 1999; Guo et al., 2017). Orthogonal approaches have been proposed where trust scores and other measures for out-of-distribution (OOD) detection are derived, typically also based on trained networks (Liang et al., 2018; Jiang et al., 2018; Papernot & McDaniel, 2018); however these latter approaches are designed to detect only truly OOD samples and do not consider the continuum of domain shifts from in-domain to truly OOD. Alternative avenues towards intrinsically uncertainty-aware networks have been followed by training probabilistic models. In particular, a lot of research effort has been put into training Bayesian neural networks, where typically a prior distribution over the weights is specified and, given the training data, a posterior distribution over the weights is inferred. This distribution can then be used to quantify predictive uncertainty. Since exact inference is untractable, a range of approaches for approximate inference has

been proposed.In particular approaches based on variational approximations have recently received a lot of attention and range from estimators of the fully factorized posterior (Blundell et al., 2015), to the interpretation of Gaussian dropout as performing approximate Bayesian inference (Gal & Ghahramani, 2016) and facilitating a complex posterior using normalising flows (Louizos & Welling, 2017). Since such Bayesian approaches often come at a high computational cost, alternative non-Bayesian approaches have been proposed, that can also account for predictive uncertainty. These include ensemble approaches, where smooth predictive estimates can be obtained by training ensembles of neural networks using adversarial examples (Lakshminarayanan et al., 2017), and evidential deep learning, where predictions of a neural net are modelled as subjective opinions by placing a Dirichlet distribution on the class probabilities (Sensoy et al., 2018). Both for Bayesian and non-Bayesian approaches, uncertainty-awareness and the quality of predictive uncertainty are typically evaluated by analysing the behaviour of the predictive entropy for out-of-domain predictions in form of gradual perturbations (e.g. rotation of an image), adversarial examples or held-out classes. However, while an increasing predictive entropy for increasingly strong perturbations can be an indicator for uncertainty-awareness, simply high predictive entropy is not sufficient for trustworthy predictions. Models can only be trusted if the confidence of predictions is calibrated, that is if the entropy matches the actual accuracy of the model. For example, if the entropy is too high, the model will yield under-confident predictions and similarly, if the entropy is too low, predictions will be over-confident. Notably, the focus of related work introduced above has been on image data and it remains unclear how these approaches perform for other data modalities, in particular when modelling sequences with long-range dependencies using complex architectures such as LSTMs (Hochreiter & Schmidhuber, 1997) or GRUs (Cho et al., 2014).

Here, we propose an efficient yet general modelling approach for obtaining calibrated, trustworthy probabilities for both in-domain samples as well as under domain shift that can readily be applied to a wide range of data modalities and model architectures. More specifically, we first introduce a simple loss function to encourage high entropy on wrong predictions and combine this with an adversarial calibration loss term. Since in practical applications it is a priori not clear what type or magnitude of domain drift will occur, we evaluate calibration under doamin drift for 10 different perturbations and 10 different noise levels not seen during training.

Our contribution in this paper is three-fold. (i) we illustrate the limitations of entropy as measure for trustworthy predictions and motivate the use of the expected calibration error for quantifying technical robustness (Dawid, 1982; DeGroot & Fienberg, 1983; Niculescu-Mizil & Caruana, 2005; Naeini et al., 2015; Guo et al., 2017). (ii) we introduce a new training strategy combining an entropy-encouraging loss with an adversarial calibration loss term and demonstrate that this results in better calibration and technical trustworthiness of predictions for diverse types of out-of-domain samples and perturbations, compared to the state-of-the-art. (iii) We apply the concept of uncertainty-awareness and trustworthiness to sequence models and demonstrate that our approach substantially improves predictive uncertainty over existing approaches when classifying long sequences. While previous studies only compared predictive entropy for one simple architecture (LeNet) and typically one type of domain shift (Sensoy et al., 2018; Louizos & Welling, 2017), we here present an extensive comparison of 4 different architectures across 10 different perturbation strategies.

## 2 TOWARDS TECHNICALLY TRUSTWORTHY PREDICTIONS

### 2.1 LIMITATIONS OF ENTROPY AS MEASURE FOR UNCERTAINTY-AWARENESS

Recent efforts in terms of evaluating predictive uncertainty have focused on entropy as measure for uncertainty-awareness for predictions under domain shift. While entropy quantifies the uncertainty encoded in the model output, it is not clear what absolute entropy is required for a model to be reliable, given a set of samples from an out-of-domain distribution. For example, a popular evaluation strategy consists of computing the absolute entropy for out-of-domain samples generated using perturbation strategies based on the images in the test set (e.g. gradual rotation of images) (Sensoy et al., 2018; Louizos & Welling, 2017). In this case, the entropy should increase with rotation angle, as the accuracy decreases in a coordinated fashion (since the model was not trained with rotated images) (Fig. 1). However, such evaluations alone are not sufficient to determine whether model predictions are technically reliable (or trustworthy), since it is not clear whether accuracy and model confidence/uncertainty are coupled in a meaningful way. Building on prior work utilising the concept of calibration for in-domain predictions, this coupling can be quantified using reliability diagrams

(Guo et al., 2017), where the model confidence (i.e. the probability associated with the predicted class label) is linked to accuracy in a stratified manner. That is, for a given set of samples obtained under domain shift, confidence and accuracy should match for all confidence levels between $1/n_{\text{classes}}$ and 1.0. For example, for the subset of samples with confidence between e.g. 60% and 70% the average accuracy should lie in that same range; this relationship should hold for all intervals. Figure 1 illustrates that the accuracy decreases, while the entropy increases if perturbed images are fed to a trained neural network (top right); however, additional information directly linking the uncertainty or confidence of a model to its accuracy is required to establish whether predictions are calibrated. This is illustrated by reliability diagrams in figure 1 (bottom row), showing accuracy as function of binned confidence and the expected calibration error (ECE) curve, summarizing the calibration gap by covering the entire spectrum of domain shifts. (DeGroot & Fienberg, 1983; Niculescu-Mizil & Caruana, 2005).

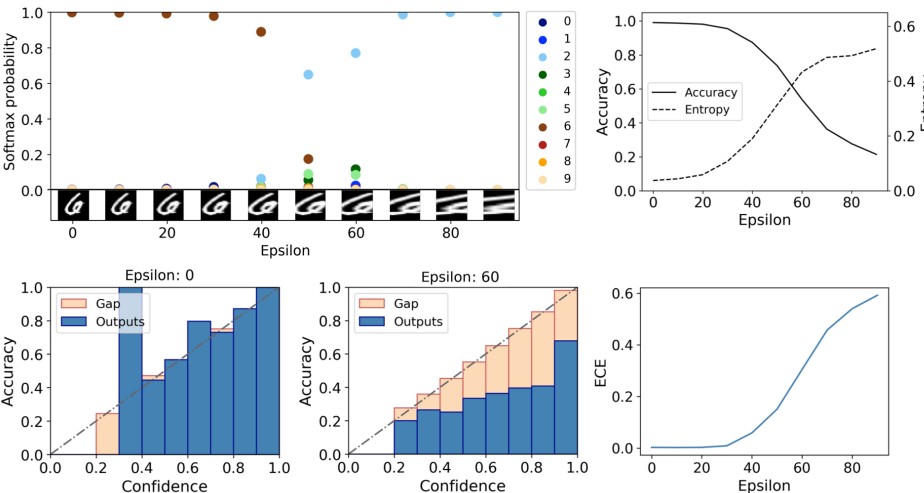

Figure 1: Calibration of the predictive uncertainty under domain shift. Here, a LeNet model is trained on MNIST data and calibration of the predictive uncertainty is evaluated on images perturbed with increasing y-zoom. Epsilon denotes the relative perturbation strength. **Top**: For in-domain samples the model has a high accuracy and low entropy, for higher domain shifts wrong predictions are often made with high confidence (*left*). While increasing domain shift results in a decreased accuracy and increased entropy, it is not clear whether this increased entropy reflects a well calibrated model confidence (*right*). **Bottom:** Only reliability diagrams and the expected calibration error (ECE) reveal that the decline in accuracy does not match the confidence of the model. *Left:* Confidence matches accuracy for most bins. *Middle:* Model makes overconfident predictions (red bars illustrate calibration gap). *Right:* ECE curve quantifies how miss-calibration changes with increasing perturbation strength.

### 2.1.1 Quantifying calibration under domain shift using the ECE curve

Let $X \in \mathbb{R}^D$ and $Y \in \{1, \dots, C\}$ be random variables that denote the $D$-dimensional input and labels in a classification task with $C$ classes, respectively. Let $h(X) = (\hat{Y}, \hat{P})$ be the output of a neural network classifier $h$ predicting a class $\hat{Y}$ and associated confidence $\hat{P}$ based on $X$. We follow Guo et al. (2017) and formally define perfect calibration such that confidence and accuracy match for all confidence levels:

$$\mathbb{P}(\hat{Y} = Y | \hat{P} = p) = p, \quad \forall p \in [0, 1] \tag{1}$$

This directly leads to a definition of miss-calibration mca as the difference in expectation between confidence and accuracy, that is

$$\text{mca} = \mathbb{E}_{\hat{P}}\left[\left|\mathbb{P}(\hat{Y} = Y | \hat{P} = p) - p\right|\right] \tag{2}$$

mca can be estimated from finite samples, by partitioning predictions into $M$ equally-spaced bins and computing a weighted average of the bins' difference between accuracy and confidence. The

resulting measure is the expected calibration error (ECE) (Naeini et al., 2015):

$$\text{ECE} = \sum_{m=1}^{M} \frac{|B_m|}{n} \big| \text{acc}(B_m) - \text{conf}(B_m) \big| \tag{3}$$

with $B_m$ being the set of indices of samples whose prediction confidence falls into its associated interval $I_m$. $\text{conf}(B_m)$ and $\text{acc}(B_m)$ are the average confidence and accuracy associated to $B_m$ respectively and $n$ the number of samples in the dataset.

It can be shown that ECE is directly connected to miss-calibration as ECE using $M$ bins converges to the M-term Riemann-Stieltjes sum of eq. 2 Guo et al. (2017). Finally, ECE can also be interpreted as a summary measures of reliability diagrams since it quantifies the calibration gap (red bars in figure 1).

ECE was recently popularised when used quantify in-domain calibration for modern neural networks Guo et al. (2017). Here, we use an ECE based measure to evaluate the calibration of a predictive model under domain drift. Since the type of domain drift that may occur after training is generally not known a priori, we define a range of distinct perturbation types not seen during training. Each perturbation strategy mimics a scenario where the data a deployed model encounters stems from a distribution that gradually shifts away from the training distribution in a different manner. For each perturbation type we compute the ECE for a range of perturbation strengths. We then generate a ECE-perturbation curve and summarize overall calibration by computing a micro-averaged ECE across all perturbation strengths.

ECE curves and micro-averaged ECE are primarily designed to evaluate calibration in domain-drift scenarios where the distribution the inputs are drawn from changes, but the set of labels remains unchanged. For truly OOD samples with labels that are not part of the train classes, entropy-based measure are a useful complementary approach to evaluate predictive uncertainty.

## 2.2 A SIMPLE APPROACH FOR CALIBRATED PREDICTIVE UNCERTAINTY ESTIMATION

### 2.2.1 PREDICTIVE ENTROPY

To mitigate overconfident predictions displayed by conventional deep neural networks, we first introduce a loss term encouraging a uniform distribution of the scores in case the model "does not know". That is, we distribute the probability mass of false predictions uniformly over $C$ classes:

$$L_S = \sum_{i=1}^{n} \sum_{j=1}^{C} -\frac{1}{C} \log(p_{ij}(1 - y_{ij}) + y_{ij}), \tag{4}$$

with $p_{ij}$ being the confidence associated to the $j$th class of sample $i$, $y_{ij}$ its one-hot encoded label. This simple loss term increases uncertainty-awareness by encouraging an increased entropy ($S$) in the presence of high predictive uncertainty without directly affecting reconstruction loss (categorical cross-entropy) and thus accuracy. This has the advantage that our approach - in contrast to state-of-the-art Bayesian neural networks such as those based on multiplicative normalizing flows or evidential deep learning - can be readily applied to complex architectures based on LSTMs or GRUs. In addition, the loss term is parameter free and thus does not require hyperparameter tuning, again facilitating easy usage.

### 2.2.2 ADVERSARIAL CALIBRATION

While the entropy-based loss term does encourage uncertainty-awareness, we found that it is beneficial to introduce an additional loss term addressing model calibration directly. Explicitly encouraging calibration for out-of-domain samples, however - e.g. via an ECE-based measure - requires knowledge on the type of perturbed or erroneous samples the model is expected to encounter. In many real-world applications it is not clear from which distribution these samples will be drawn and, for model predictions to be truly trustworthy requires robustness against all such potential out-of-domain samples. That is, we would like our model to be technically robust for inputs around an $\epsilon$-neighbourhood of the in-domain training samples, for a wide range of $\epsilon$ and for all $2^D$ directions in $\{-1, 1\}^D$. While inputs from a random direction are unlikely to be representative examples for generic out-of-domain samples, by definition adversarial examples are generated along a dimension where the loss is high. Lakshminarayanan et al. (2017) show that adversarial training can improve the smoothness of predictions, in particular when training an ensemble of 5 neural networks in an adversarial fashion. Here, we demonstrate that using adversarial samples to directly optimise model

calibration (rather than the squared error of one-hot encoded labels (Lakshminarayanan et al., 2017)) results in substantially more trustworthy predictions for out-of-domain samples from a large number of unrelated directions.

We implement an ECE-inspired calibration loss by minimizing miss-calibration for samples generated using the fast gradient sign method (FGSM) (Goodfellow et al., 2014), with $\epsilon$ ranging from 0 to 0.5 (sampled at 10 equally spaced bins at random). To this end we minimise the L2 norm of the difference between the predicted confidence of a sample $i$, which we denote as $\mathrm{conf}(i)$, and its corresponding binned accuracy $\mathrm{acc}(B_{m_i})$, for all samples. This is directly motivated by the definition of ECE (eq. 2.1.1) and as for the computation of ECE, we partition the predictions of the network into $M$ equally-spaced bins, with $m_i \in \{1, \ldots, M\}$ being bin into which sample $i$ falls. As for ECE, $B_{m_i}$ is the set of indices of samples falling in bin $m_i$ and $\mathrm{acc}(B_{m_i})$ the average accuracy of samples $B_{m_i}$. We set $M = 10$ for all experiments.

$$L_{\mathrm{adv}} \quad = \quad \sqrt{\sum_{i=1}^{n} \left(\mathrm{acc}(B_{m_i}) - \mathrm{conf}(i)\right)^2} \tag{5}$$

The final loss balancing a standard reconstruction loss (categorical cross entropy (CCE)) against the entropy and adversarial calibration loss can then be written as $L = L_{\mathrm{CCE}} + \lambda_{\mathrm{adv}} L_{\mathrm{adv}} + \lambda_S L_S$.

The choice of hyperparameters $\lambda_{\mathrm{adv}}$ and $\lambda_S$ is described in the appendix along with a robustness analysis, an ablation study for both loss terms and a summary of the algorithm. Note that we do not use the FGSM samples for adversarial training in the sense that we do not try to minimize the reconstruction error (cross entropy) for those samples.

## 3 Experimental results

We compare our approach for fast adversarial calibration to both Bayesian and non-Bayesian work and perform an extensive set of experiments. We evaluate model trustworthiness by quantifying model calibration for 10 distinct strategies to generate out-of-domain samples. We show that our approach is able to yield technically trustworthy predictions across 4 datasets, 4 model architectures and three data modalities. We assess 9 distinct image-based perturbation types including left rotation, right rotation, shift in x direction, shift in y direction, xy shift, shear, zoom in x direction, zoom in y direction and xy zoom for image data. In addition, we investigate robustness to random word swaps for text data. More specifically, a perturbation is generated by first drawing a random set of words in a corpus. Next each of these words is replaced by a word drawn at random from the vocabulary. For all perturbation strategies, perturbed samples were generated at 10 different levels, starting at no perturbation, until accuracy reached random levels; relative perturbation strength is denoted by epsilon. The micro-averaged ECE for a specific perturbation strategy was computed by first perturbing each sample in the test set at 10 different levels and then calculating the overall ECE across all samples. By computing this micro-averaged ECE for 100 different perturbation scenarios, we quantify the ability of neural networks to to yield well-calibrated, technically robust predictions in diverse circumstances.

We first show that our modelling approach substantially outperforms existing approaches for sequence models and then illustrate improved performance for image data. To evaluate our modelling approach for sequence data, we fit models on the following datasets:

1. Sequential MNIST. 10 classes of handwritten digits. Images are converted to pixel-wise sequences of length 28x28.

2. 20 Newsgroups. News articles partitioned into 20 classes. News classes are modelled as sequences of words using word embeddings. We used the 20,000 most common words as vocabulary and a maximum word length of 2500.

We fitted LSTM and GRU models with one hidden layer for all sequence modelling tasks.

For the image classification tasks, we fitted a LeNet model to MNIST data in order to establish a fair comparison to the state-of-the-art (Guo et al., 2017; Sensoy et al., 2018). To evaluate the performance for more complex architectures, we further fitted a deep neural net with VGG19 architecture on the CIFAR10 dataset. We used standard splits into training and test set for all datasets.

We compared the following modelling approaches: (i) *L2-Dropout*, referring to a standard neural

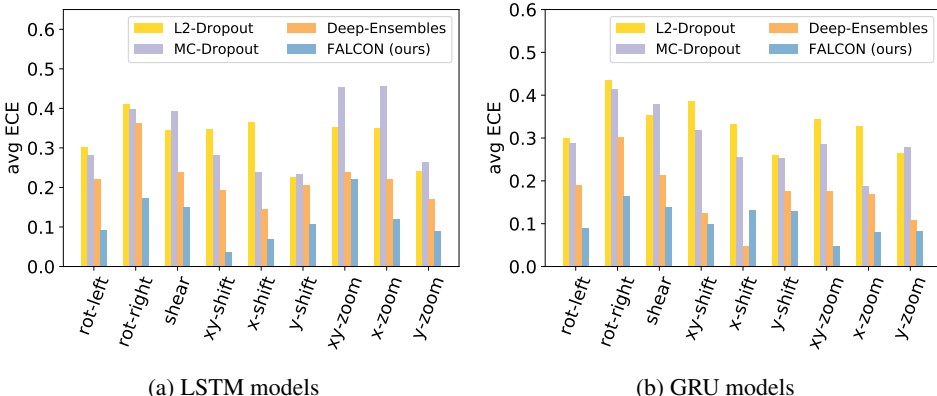

Figure 2: Technical robustness of sequence models for classifying sequential MNIST data, quantified by computing the micro-averaged expected calibration error (lower is better). FALCON results in consistently well calibrated and robust predictions across 9 different perturbation strategies with substantially lower micro-averaged ECEs compared to existing methods, both for LSTM and GRU models. For fair comparison, we only show micro-averaged ECE for models with competitive accuracy, omitting EDL

net with L2 regularisation as baseline, (ii) *MC-Dropout* corresponding to the modelling approach presented by Gal & Ghahramani (2016), (iii) *Deep Ensembles* referring to an approach based on an ensemble of neural nets trained using adversarial examples (Lakshminarayanan et al., 2017), (iv) *EDL* referring to Evidential Deep Learning (Sensoy et al., 2018), (v) *MNF* referring to a Bayesian neural network trained using multiplicative normalising flows (Louizos & Welling, 2017) and (vi) *FALCON*, which is our method based on Fast AdversariaL CalibratiON. Additional comparisons to temperature scaling (Guo et al., 2017) and stochasctic variational inference (SVI) based on Flipout (Wen et al., 2018) are shown in the Appendix A.5

### 3.1 PREDICTIVE UNCERTAINTY FOR SEQUENCE MODELING

We trained LSTM models with one hidden layer of 130 hidden units using the RMSPROP optimizer. GRU models were trained with one hidden layer of 250 hidden units to reflect the reduced complexity of GRU cells compared to LSTM cells. The Bayesian neural network based on multiplicative normalizing flows (MNF) was developed for convolutional neural networks; since the transfer of such a complex modelling approach from convolutional neural networks to recurrent neural networks is out of the scope of this work, we omitted MNF in our comparison of sequence models.

**Sequential MNIST**  For deep ensembles of LSTMs trained on sequential MNIST we found that models did not converge when training the networks with adversarial examples; we therefore also trained ensembles with a reduced $\epsilon$ of 0.005 and report performance for this modified Deep Ensemble approach. For the deep ensemble of GRUs on sequential MNIST and the deep ensemble of LSTMs on the 20 Newsgroups data, we report performance with standard adversarial training ($\epsilon = 0.01$). Fitting LSTM models on sequential MNIST is a challenging task (Bai et al., 2018), and it was only possible to achieve state-of-the-art predictive power with EDL for shorter sequences (downsampling of images before conversion to sequence). While performance of GRUs was better for all modelling approaches, EDL also did not achieve a competitive accuracy (Table S2 in appendix A.7). We found that our approach achieved competitive predictive power for LSTM and GRU models and substantially improved calibration of the predictive uncertainty for both models (Figure 2, Table 1). This illustrates that in contrast to existing approaches FALCON is able to yield well-calibrated and trustworthy predictions without compromising on accuracy, even for challenging tasks such as classifying long sequences with LSTMs.

**20 Newsgroups**  To further evaluate the ability of FALCON to model sequence data, we compared the performance of FALCON to existing approaches for an NLP task. To this end, we trained LSTMs to classify news articles into one of 20 classes. We generated vector representations of words using

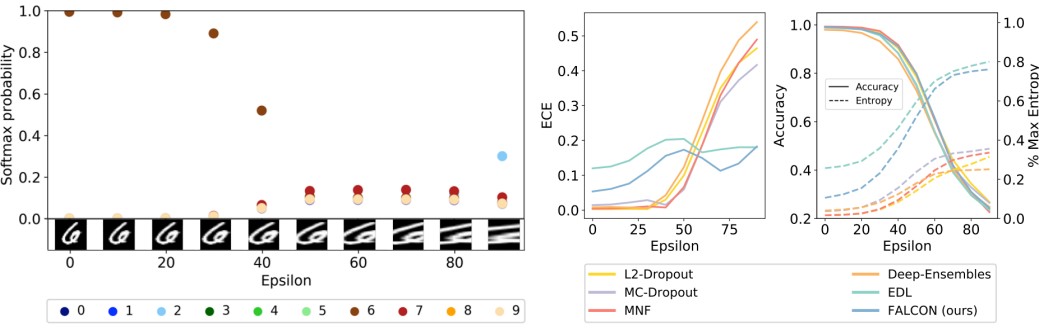

Figure 4: Calibration of the predictive uncertainty under domain shift generated by increasing the y-zoom of each image in the test set in 10 steps (MNIST data). **Left** With increasing domain shift the confidence of predictions with FALCON decreases such that they match accuracy (c.f. overconfident predictions of same samples with L2 in Fig. 1). **Middle**: expected calibration error at 10 increasingly large levels of y-zoom. Only EDL and FALCON maintain a low ECE across all levels of y-zoom. **Right:** Entropy increases with larger y-zoom for all methods. While EDL starts at the highest entropy, this reflects under-confident predictions for low levels of perturbation (c.f. high ECE in middle panel, figure S3 (appendix)). Accuracy decreases with larger zoom to almost random levels.

the pre-trained GLOVE embedding (length 100) and used the first 2500 words of an article as input for an LSTM. We trained LSTMs with one hidden layer of 130 hidden units and evaluated it on a perturbation strategy based on random word swaps. To establish a perturbation strategy with gradually increasing perturbations, we varied the fraction of words drawn from each sample between 0% and 45% in 5% steps (gradually decreasing accuracy to random levels).

Similar to the LSTM model trained on sequential MNIST, we found that EDL did not achieve competitive predictive power, with an accuracy of 49.3% only. In contrast, FALCON resulted in well-calibrated predictions while maintaining a competitive accuracy of 75.7%, compared to 75.9%, 72.8% and 77.3% for L2-Dropout, MC-Dropout and Deep Ensemble respectively. As before, the model confidence of FALCON was substantially better calibrated than existing methods (Figure 3).

## 3.2 PREDICTIVE UNCERTAINTY FOR IMAGE CLASSIFICATION

We next evaluated the trustworthiness of predictions for image classification tasks. To establish a fair comparison with state-of-the-art models, including Bayesian neural networks, we first trained the 5 existing approaches and evaluated them on 9 different perturbation strategies (not used during training). While with increasingly strong perturbations the predictive entropy increased for all models, this was not necessarily matched by a good calibration across the range of the perturbation. At the typical example of the perturbation y-zoom, it becomes clear that for most methods entropy did not increase sufficiently fast to match the decrease in accuracy, resulting in increasingly overconfident predictions and an increasing ECE for stronger perturbations (Fig. 4). While FALCON and EDL yielded well-calibrated predictions that were robust across all perturbation levels, it is worth noting that EDL has a substantially higher ECE for in-domain predictions, reflecting under-confident predictions on the test set (see also Fig. S4, S5, appendix). We observed this tendency of EDL towards under-confidence when faced with new samples drawn from the same distribution as the training data (known unknowns) also for a different dataset and architecture (VGG19 on CI-

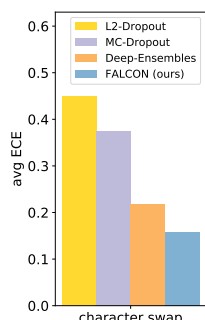

Figure 3: Expected calibration error for 20 Newsgroups data.

FAR10; $ECE_{FALCON} = 0.107$, $ECE_{EDL} = 0.125$ on the test set). While predictions with FALCON were also slightly under-confident for low levels of perturbation, this has arguably a smaller practical relevance than the substantial overconfidence of other baseline methods (c.f appendix A.6 for a more in-depth analysis). We observed a similar behaviour across all other 8 perturbation strategies, which was reflected in the lowest micro-averaged ECE for FALCON, followed by EDL (Figure 5; Table 1).

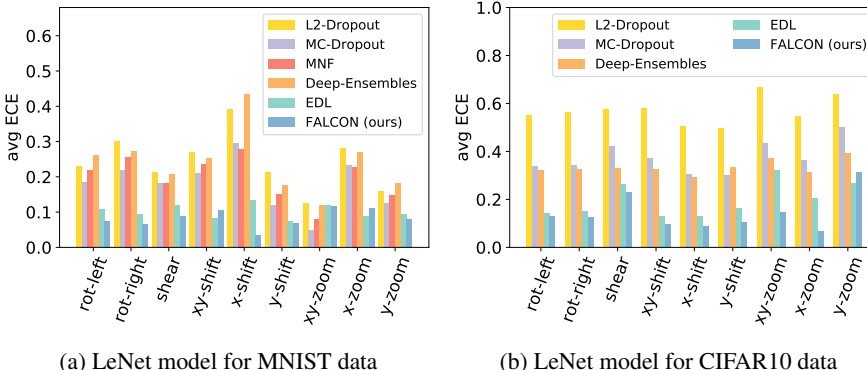

(a) LeNet model for MNIST data      (b) LeNet model for CIFAR10 data

Figure 5: Technical robustness of image classification models, quantified by computing the micro-averaged expected calibration error (lower is better). FALCON results in consistently well calibrated and robust predictions across 9 different perturbation strategies.

To evaluate the technical robustness and calibration of FALCON on a more complex architecture for image classification, we trained a VGG19 model on the CIFAR10 dataset. We again observed a similar trend as for the MNIST data, with FALCON yielding well calibrated predictions across all perturbation strategies (Figure 5). The considerable overconfidence of most baseline methods (except EDL) when making predictions on OOD samples (epsilon 90) is also reflected by a substantially lower maximum entropy (Fig. S7, Appendix A.7) and the overall distribution of confidence scores (Fig. S8). Note that we omitted MNF due to the large memory requirements stemming from the use of multiplicative normalising flows.

Table 1: Test accuracy and mean ECE across all 9 perturbation strategies for the LeNet model trained on MNIST and the VGG19 model trained on CIFAR10

|  | LeNet-MNIST | | VGG19-CIFAR10 | |
|  | Test acc. | Mean ECE | Test acc. | Mean ECE |
| --- | --- | --- | --- | --- |
| L2-Dropout | 0.99 | 0.243 | 0.88 | 0.57 |
| MC-Dropout | 0.992 | 0.179 | 0.839 | 0.377 |
| MNF | 0.993 | 0.197 | NA | NA |
| Deep-Ensembles | 0.98 | 0.242 | 0.847 | 0.334 |
| EDL | 0.989 | 0.102 | 0.876 | 0.197 |
| FALCON | 0.991 | **0.082** | 0.871 | **0.146** |

## 4 DISCUSSION AND CONCLUSION

We presented a fast, simple and generalizable approach for encouraging well-calibrated uncertainty-awareness of deep neural networks. To this end, we combine an entropy encouraging loss-term with an adversarial calibration loss and show on diverse data modalities and model architectures that our approach yields well-calibrated predictions for both in-domain and out-of-domain samples generated based on 10 distinct perturbations. We present a detailed analysis of calibration under domain drift for recurrent neural networks and identify major drawbacks of existing methods that were developed for (and evaluated on) image classification tasks. Thus, EDL was only able to result in networks with a high accuracy when trained on short sequences; both for the sequential MNIST and and 20 Newsgroups data, the EDL approach resulted in a substantially lower accuracy compared to baseline LSTM and GRU models. While MC dropout is easy to fit and fast, it results only in small improvements over the L2-Dropout baseline, especially for sequence data. In contrast, our modeling approach is fast and robust, with well-calibrated predictive uncertainty across 10 perturbations, 4 datasets, 4 model architectures and three data modalities.

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

## A    APPENDIX

### A.1    PARAMETER AND HYPERPARAMETER SETTINGS

Deep Ensembles, MNF, and EDL were trained with default values for method-specific hyper-parameters (e.g. number of neural networks in a Deep Ensemble). In addition, the following hyperparameters were picked using hyperparameter searches. For all methods, the learning rate was chosen from $\{1e-5, 5e-5, 1e-4, 5e-4, 1e-3, 5e-3\}$. In addition, for the baseline method (L2), our method (FALCON), Deep Ensembles and EDL, dropout was chosen from $\{0, 0.5\}$ and L2-regularisation from $\{0.0, 0.001, 0.005, 0.01, 0.05\}$. For EDL we chose the KL regularisation from $\{0.5, 1., 5., 15., 10., 30., 50.\}$. For a fair comparison with this state-of-the art model, we chose $\lambda_S$ from this same set of values for FALCON and $\lambda_{adv}$ from $\{0.25, 1e-1, 1e-2, 1e-3, 1e-4\}$. We also assessed the effect of the individual loss terms $\lambda_{adv}$ and $\lambda_S$ and the robustness of FALCON with respect to the choice of $\lambda_S$ and $\lambda_{adv}$ (section A.4).
We used a batch size of 128 for all models and standard splits in train and test data for all datasets.

For the 20 Newsgroups dataset we used the keras tokenizer to format text samples, converting words into lower case, removing punctuation and special characters `!"#$%&()*+,-./:;<=>?@[\\]^_'{}~\t\n'`.

### A.2    PERTURBATION STRATEGIES

In practice it is not clear what type of perturbation a model may encounter. To assess how neural networks cope in diverse settings, we generated out-of-domain samples based on 10 different perturbation strategies. Each perturbation strategy mimics a scenario where the data a deployed model encounters stems from a distribution that gradually shifts away from the training distribution in a different manner. Samples generated with maximum perturbation strength correspond for example to corrupted or erroneous samples a deployed model may face, unperturbed samples correspond to those drawn from the same distribution as the training data ("known unknowns"). Trustworthy AI models should yield well-calibrated confidence scores in all those settings that it may encounter throughout its life-cycle. We quantify this based on the expected calibration error, micro-averaged across all perturbation strengths, including no perturbation (Tables S6-S10).
For all perturbation strategies we chose 10 levels of perturbation, starting at no perturbation, such that accuracy levels were close to random for maximum perturbation strength (Table S5). Specific levels of perturbation are listed in Table S4; for visualisation purposes we re-scaled all perturbation-specific parameters to range from 0 to 90 (in steps of 10) and denote this general perturbation strength as epsilon. Perturbations include image transformations (rotation, shift, zoom, shear) as well as a word perturbation (word swap). For sequential MNIST, perturbations were performed on the image before transforming the image to a sequence.

## A.3 TRAINING ALGORITHM

Training was performed following Algorithm 1, summarizing the description in section 2.2.

---

**Algorithm 1** FALCON with set of perturbation levels
$\mathcal{E} = \{0, 0.05, 0.1, 0.15, 0.2, 0.25, 0.3, 0.35, 0.4, 0.45\}$ (n.b. $\epsilon = 0$ encourages
in-domain calibration) , mini batch size $b$, number of ECE bins $M$ and training set $(X, Y)$.
$m_i \in \{1, \ldots, M\}$ denotes the bin into which sample $i$ falls and $B_{m_i}$ is the set of indices of
samples falling in bin $m_i$. $\text{acc}(B_{m_i})$ the average accuracy of samples $B_{m_i}$.

---

1: **repeat**
2:      Read minibatch $MB = (\{X_1, \ldots, X_b\}, \{Y_1, \ldots, Y_b\})$ from training set
3:      Randomly sample $\epsilon_{MB}$ from $\mathcal{E}$
4:      Generate FGSM minibatch $MB_{adv}$ of size $b$ from samples in $MB$ using $\epsilon_{MB}$
5:      Compute $L_{CCE}$ and $L_S$ and do one training step using mini batch $MB$
6:      Compute predictions for all samples in $MB$ and partition into $M$ equally spaced bins
7:      Compute binned accuracy $\text{acc}(B_{m_i})$ for all samples $i$ in $MB_{adv}$
8:      Compute $L_{adv}$ based on $MB_{adv}$ and do one training step using $MB_{adv}$
9: **until** training converged

---

## A.4 ABLATION STUDY AND SENSITIVITY ANALYSIS

In order to investigate the influence of the individual loss terms on calibration, we performed an ablation study, omitting one of the two loss terms, $L_S$ and $L_{adv}$ respectively. While either loss term results in an improved calibration compared to the L2-dropout baseline, combining both terms yields consistently better results.

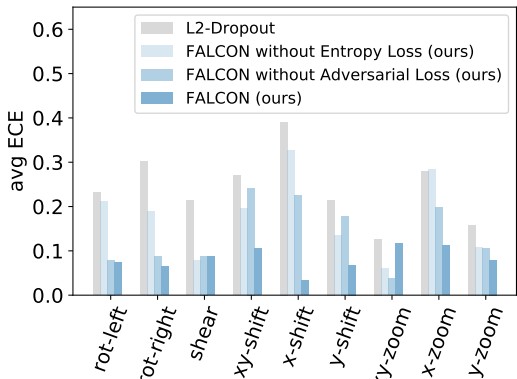

Figure S1: Micro-averaged ECE for FALCON with only one loss term loss term, based on the LeNet model trained on MNIST.

We further performed a robustness analysis in order to quantify the dependence of model performance on calibration under domain drift. To this end, we trained LeNet models on MNIST varying both hyperparameters over wide ranges. We first fixed $\lambda_S$ at the optimal value of 50 and varied $\lambda_{adv}$ between 0.0005 and 0.03 and computed the micro-averaged ECE for perturbation yzoom for all hyperparameter combinations. Next, we fixed $\lambda_{adv}$ at the optimal value of 0.02 and varied $\lambda_S$ between 10 and 100. We found that even when varying both hyperparameters over a wide range, ECE remained robust and varied by less than 0.04 for $\lambda_{adv}$ and less than 0.06 for $\lambda_S$. Accuracy was not affected by the choice of either $\lambda$ and remained between 0.985 and 0.991.

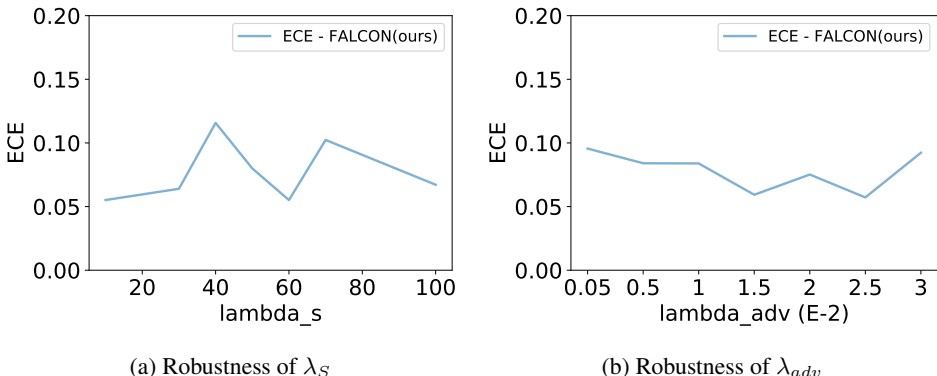

(a) Robustness of $\lambda_S$          (b) Robustness of $\lambda_{adv}$

Figure S2: Robustness of hyperparameters. Even when varying both hyperparameters, micro-averged ECE for perturbation yzoom was robust.

### A.5 ADDITIONAL BASELINES

In addition to the 5 baselines evaluated in depth in the main text, we also assessed two additional baselines, namely SVI based on Flipout and temperature scaling. We found that temperature scaling, while improving in-domain calibration, is not beneficial for calibration under domain-drift (Fig. S3). In addition, we observe that SVI results in a decreased accuracy compared to other baselines and while calibration under domain drift improves in comparison to L2-dropout, it is substantially worse than FALCON and also EDL (Table S1). Finally, we observe that FALCON has a substantially better performance than both SVI and temperature scaling for truly OOD samples (corresponding to strongest perturbations), which is reflected in a higher entropy and more overconfident confidence scores (Figs. S7, S8.)

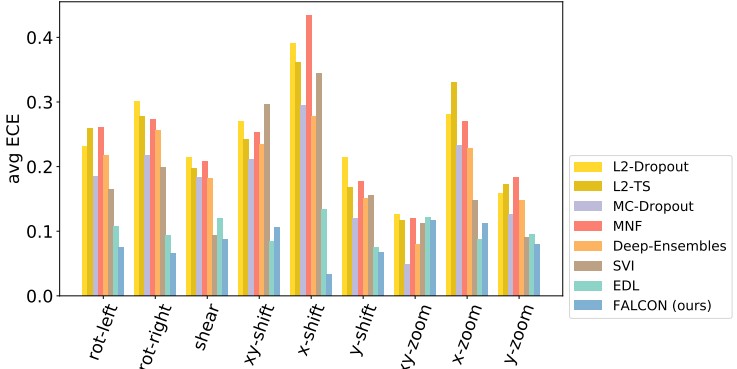

Figure S3: Calibration under domain shift of the LeNet model for MNIST data, quantified by computing the micro- averaged expected calibration error (lower is better). FALCON results in consistently well calibrated and robust predictions across 9 different perturbation strategies. L2-TS stands for temperature scaling, SVI stands for Stochastic Variational Inference.

Table S1: Test accuracy and mean ECE across all 9 perturbation strategies for the LeNet model trained on MNIST, including additional baselines L2-TS and SVI.

|  | Test acc. | Mean ECE |
| --- | --- | --- |
| L2-Dropout | 0.99 | 0.243 |
| MC-Dropout | 0.992 | 0.179 |
| MNF | 0.993 | 0.197 |
| Deep-Ensembles | 0.98 | 0.242 |
| EDL | 0.989 | 0.102 |
| L2-TS | 0.99 | 0.236 |
| SVI | 0.974 | 0.176 |
| FALCON | 0.991 | **0.082** |

### A.6 UNDER-CONFIDENCE OF FALCON AND EDL FOR SMALL PERTURBATIONS

We observe e.g. in Fig. 4 (middle), that FALCON and EDL have a tendency to make slightly under-confident predictions for small perturbations. To further investigate the extend and practical relevance of this under-confidence, we generated an empirical CDF of the confidence scores for all approaches. This confirms the issue, but also illustrates that FALCON is consistently less under-confident than EDL: While for FALCON 13.3% of predictions on the test set are made with a confidence of less than 90%, this increases to 30.5% for EDL, which is substantially higher than the 7.7% we observe for the well-calibrated Deep Ensembles. In other words, only a minority of 5.6% of all predictions made by FALCON are made with a confidence of less than 90% and should have been made with a higher confidence. This can also be seen at the level of individual samples, as illustrated in figure

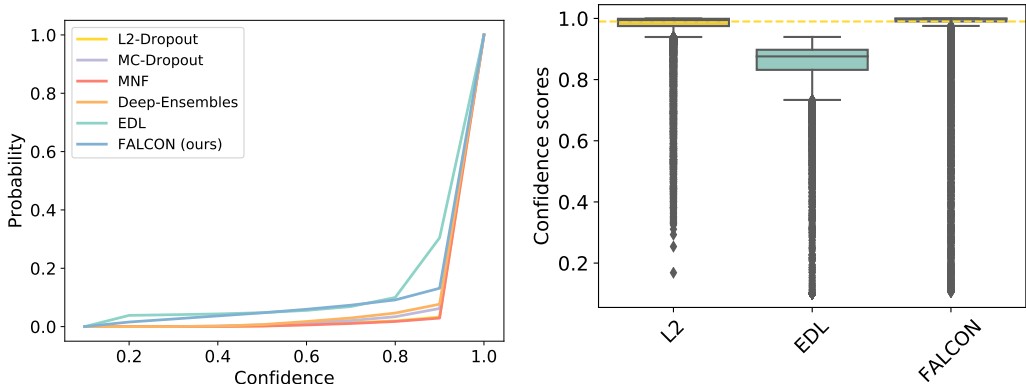

(a) Empirical CDF of confidence levels for all approaches.

(b) Boxplots across all confidence levels. The dashed line indicates the accuracy (L2-dropout).

Figure S4: Distribution of confidence levels for no peturbation . EDL makes the most underconfident predictions, which can be seen both in the boxplot as well as in the empirical CDF. While FALCON also makes underconfident predictions, this is notably less severe than for EDL.

S5. EDL makes noticeably under-confident predictions for small domain shifts before the entropy increases and confidence scores match accuracy. While FALCON also makes slightly under-confident predictions for in-domain samples, the corresponding confidence scores are still substantially closer to 1 (matching the near perfect test accuracies for MNIST). Like EDL, FALCON does not make over-confident predictions when moving further away from the training domain (epsilon greater 40). This substantial over-confidence e.g. illustrated in figure S8 has arguably a much bigger effect on a decision making process than the slight under-confidence of FALCON affecting only a small fraction of samples: an underconfident prediction of 89% rather than well-calibrated 95% is unlikely to affect a decision making process in a real world application to the same extent as an overconfidence of 80% rather than well-calibrated 10% (maximum entropy).

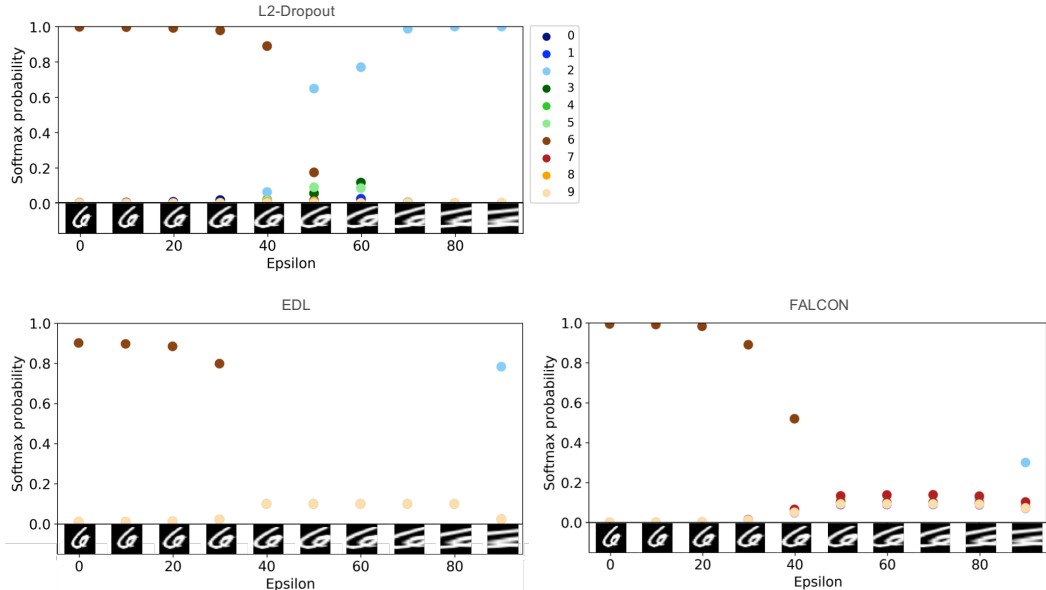

Figure S5: Softmax probabilities of a test sample with increasingly strong perturbation (y-zoom; same sample series as Fig. 1 and Fig. 4). **Top:** Predictions of L2-Dropout model start with a very high confidence, corresponding to a good calibration (Fig. 4 Middle), however, for strong perturbations (epsilon greater than 40) false predictions are made with a very high confidence, reflecting the typical overconfident behaviour of the L2-Dropout model when moving away from in-domain samples. **Bottom:** EDL (*left*) makes noticeably under-confident predictions for small domain shifts before the entropy increases and confidence scores match accuracy. While FALCON (*right*) also makes slightly under-confident predictions for in-domain samples, the corresponding confidence scores are still substantially closer to 1. Like EDL, FALCON does not make over-confident predictions when moving further away from the training domain (epsilon greater 40).

## A.7 ADDITIONAL FIGURES AND TABLES

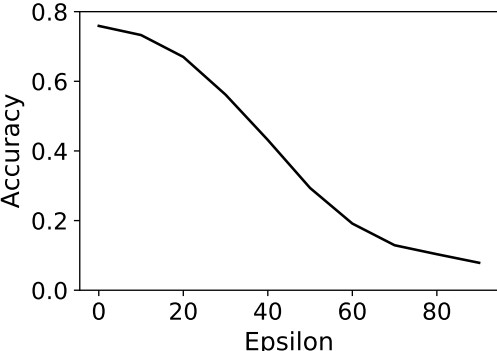

Figure S6: Test accuracy for the L2-Dropout model trained on the 20 Newsgroups data. Accuracy declines gradually with increasing fraction of swapped words until it reaches random levels.

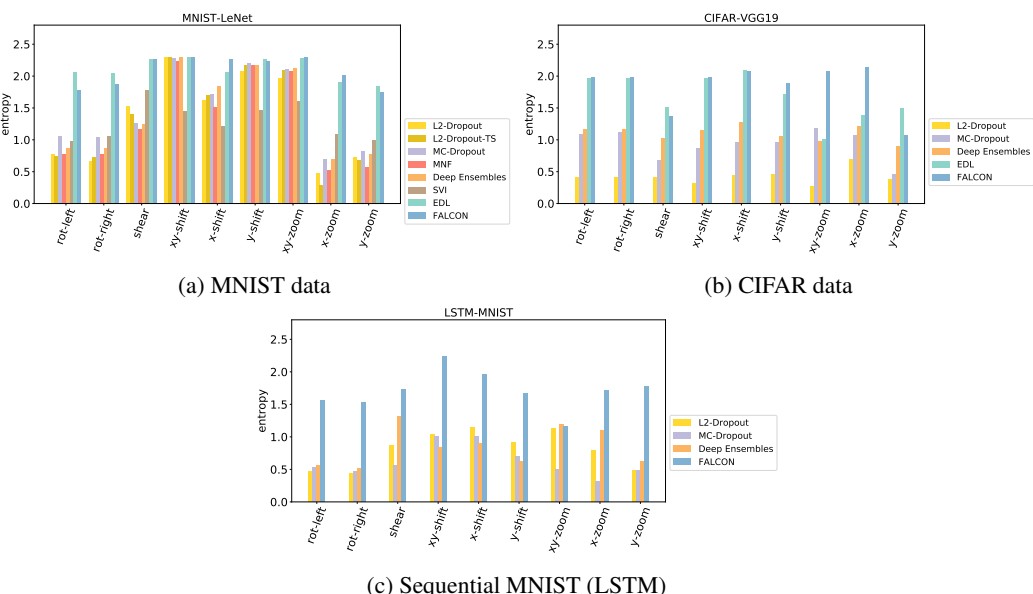

(a) MNIST data

(b) CIFAR data

(c) Sequential MNIST (LSTM)

Figure S7: Entropy for OOD predictions based on epsilon 90 on the y-zoom perturbation; with entropy approaching random levels, well calibrated predictions correspond to an entropy close to maximal entropy. The good calibration of FALCON and, for the MNIST data also EDL, is reflected by entropy levels that are substantially higher than for the other baseline methods.

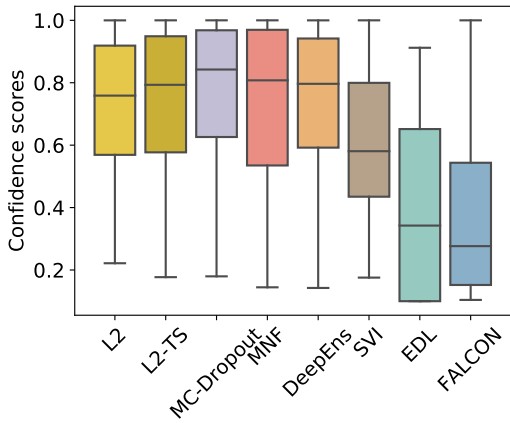

Figure S8: Distribution of confidence scores for OOD predictions based on epsilon 90 on the y-zoom perturbation (across all predictions); predictions by baseline models are often severely overconfident.

Table S2: Test accuracy and average ECE (lower is better) across all perturbation strategies for LSTM and GRU models.

| | LSTM | | GRU | |
| --- | --- | --- | --- | --- |
| | Test acc. | Mean ECE | Test acc. | Mean ECE |
| L2-Dropout | 0.986 | 0.327 | 0.991 | 0.334 |
| MC-Dropout | 0.986 | 0.334 | 0.98 | 0.296 |
| Deep-Ensemble | 0.99 | 0.222 | 0.99 | 0.168 |
| FALCON | 0.978 | **0.118** | 0.988 | **0.108** |

Table S3: Accuracy of EDL and the L2-Dropout model for downsampled images. For longer sequences EDL does not achieve competitive predictive power.

| | LSTM | | GRU | |
| Img size | L2-Drp | EDL | L2-Drp | EDL |
|---|---|---|---|---|
| 6x6 | 0.968 | 0.8203 | 0.964 | 0.9678 |
| 10x10 | 0.982 | 0.8484 | 0.987 | 0.9845 |
| 14x14 | 0.990 | 0.8223 | 0.989 | 0.9865 |
| 16x16 | 0.988 | 0.7775 | 0.990 | 0.9904 |
| 20x20 | 0.986 | 0.5513 | 0.991 | 0.9905 |
| 24x24 | 0.986 | 0.3688 | 0.989 | 0.9323 |
| 28x28 | 0.986 | 0.3907 | 0.991 | 0.8384 |

Table S4: For each perturbation we varied the perturbation-specific parameter such that it ranged from no perturbation to a maximum perturbation corresponding to an accuracy close to random. For rotation, perturbation is the (left or right) rotation angle in degrees, shift is measured in pixels in x or y direction, for shear the perturbation is measured as shear angle in counter-clockwise direction in degrees, for zoom the perturbation is zoom in x or y direction. Word swap is quantified as relative number of swapped words. Only FGSM is used during training and measured as the relative amount of noise $\epsilon$.

| Perurbation | Perturbation-specific parameter | | | | | | | | | |
|---|---|---|---|---|---|---|---|---|---|---|
| FGSM | 0 | 0.05 | 0.1 | 0.15 | 0.2 | 0.25 | 0.3 | 0.35 | 0.4 | 0.45 |
| rot left | 0 | 350 | 340 | 330 | 320 | 310 | 300 | 290 | 280 | 270 |
| rot right | 0 | 10 | 20 | 30 | 40 | 50 | 60 | 70 | 80 | 90 |
| Shear | 0 | 10 | 20 | 30 | 40 | 50 | 60 | 70 | 80 | 90 |
| xyshift | 0 | 2 | 4 | 6 | 8 | 10 | 12 | 14 | 16 | 18 |
| xshift | 0 | 2 | 4 | 6 | 8 | 10 | 12 | 14 | 16 | 18 |
| xyshift | 0 | 2 | 4 | 6 | 8 | 10 | 12 | 14 | 16 | 18 |
| xyzoom | 1 | 0.90 | 0.80 | 0.70 | 0.60 | 0.50 | 0.40 | 0.30 | 0.20 | 0.10 |
| xzoom | 1 | 0.90 | 0.80 | 0.70 | 0.60 | 0.50 | 0.40 | 0.30 | 0.20 | 0.10 |
| yzoom | 1 | 0.90 | 0.80 | 0.70 | 0.60 | 0.50 | 0.40 | 0.30 | 0.20 | 0.10 |
| word swap | 0 | 0.05 | 0.1 | 0.15 | 0.2 | 0.25 | 0.3 | 0.35 | 0.4 | 0.45 |

Table S5: Test accuracy for the L2-dropout LeNet model trained on MNIST. Accuracy is listed for no perturbation (epsilon = 0) and maximum perturbation (epsilon = 90) on the test set. For all perturbations accuracy declines to almost random levels.

| | Test Accuracy | |
| Perturbation | No perturbation | Max. perturbation |
|---|---|---|
| rot left | 0.991 | 0.19 |
| rot right | 0.991 | 0.184 |
| shear | 0.991 | 0.132 |
| xshift | 0.991 | 0.097 |
| xyshift | 0.991 | 0.095 |
| xyzoom | 0.991 | 0.087 |
| xzoom | 0.991 | 0.188 |
| yshift | 0.991 | 0.14 |
| yzoom | 0.991 | 0.242 |

Table S6: Micro-averaged ECE for LeNet model trained on MNIST

|  | rot left | rot right | shear | xyshift | xshift | yshift | xyzoom | xzoom | yzoom |
|---|---|---|---|---|---|---|---|---|---|
| L2-Dropout | 0.231 | 0.301 | 0.214 | 0.27 | 0.391 | 0.215 | 0.127 | 0.281 | 0.158 |
| MC-Dropout | 0.185 | 0.218 | 0.183 | 0.211 | 0.294 | 0.12 | 0.049 | 0.232 | 0.126 |
| MNF | 0.218 | 0.256 | 0.182 | 0.235 | 0.278 | 0.15 | 0.08 | 0.228 | 0.147 |
| Deep-Ensembles | 0.261 | 0.273 | 0.208 | 0.253 | 0.433 | 0.178 | 0.12 | 0.271 | 0.183 |
| EDL | 0.108 | 0.094 | 0.121 | **0.084** | 0.133 | 0.075 | 0.121 | **0.087** | 0.095 |
| FALCON | **0.074** | **0.065** | **0.088** | 0.106 | **0.033** | **0.068** | **0.117** | 0.113 | **0.08** |

Table S7: Micro-averaged ECE for VGG19 model trained on CIFAR10

|  | rot left | rot right | shear | xyshift | xshift | yshift | xyzoom | xzoom | yzoom |
|---|---|---|---|---|---|---|---|---|---|
| L2-Dropout | 0.551 | 0.563 | 0.576 | 0.582 | 0.507 | 0.496 | 0.669 | 0.546 | 0.639 |
| MC-Dropout | 0.339 | 0.343 | 0.423 | 0.374 | 0.307 | 0.302 | 0.436 | 0.364 | 0.502 |
| Deep-Ensembles | 0.321 | 0.326 | 0.332 | 0.325 | 0.293 | 0.333 | 0.373 | 0.314 | 0.392 |
| EDL | 0.144 | 0.15 | 0.262 | 0.132 | 0.13 | 0.164 | 0.32 | 0.206 | **0.267** |
| FALCON | **0.132** | **0.126** | **0.23** | **0.098** | **0.087** | **0.107** | **0.148** | **0.069** | 0.316 |

Table S8: Micro-averaged ECE for LSTM model trained on sequential MNIST

|  | rot left | rot right | shear | xyshift | xshift | yshift | xyzoom | xzoom | yzoom |
|---|---|---|---|---|---|---|---|---|---|
| L2-Dropout | 0.302 | 0.411 | 0.346 | 0.348 | 0.366 | 0.226 | 0.353 | 0.352 | 0.242 |
| MC-Dropout | 0.281 | 0.399 | 0.394 | 0.282 | 0.24 | 0.235 | 0.454 | 0.456 | 0.264 |
| Deep-Ensembles | 0.221 | 0.363 | 0.24 | 0.194 | 0.145 | 0.206 | 0.24 | 0.221 | 0.172 |
| FALCON | **0.092** | **0.174** | **0.15** | **0.036** | **0.069** | **0.106** | **0.221** | **0.121** | **0.09** |

Table S9: Micro-averaged ECE for GRU model trained on sequential MNIST

|  | rot left | rot right | shear | xyshift | xshift | yshift | xyzoom | xzoom | yzoom |
|---|---|---|---|---|---|---|---|---|---|
| L2-Dropout | 0.301 | 0.435 | 0.354 | 0.388 | 0.332 | 0.259 | 0.345 | 0.327 | 0.266 |
| MC-Dropout | 0.289 | 0.414 | 0.379 | 0.319 | 0.255 | 0.253 | 0.287 | 0.188 | 0.279 |
| Deep-Ensembles | 0.191 | 0.301 | 0.214 | 0.125 | **0.049** | 0.176 | 0.176 | 0.169 | 0.109 |
| FALCON (ours) | **0.09** | **0.165** | **0.14** | **0.099** | 0.132 | **0.13** | **0.049** | **0.081** | **0.083** |

Table S10: Micro-averaged ECE for LSTM model trained on 20 Newsgroups data

|  | Character swap |
|---|---|
| character swap | |
| L2-Dropout | 0.449 |
| MC-Dropout | 0.375 |
| Deep-Ensembles | 0.218 |
| FALCON (ours) | **0.158** |

