# OpenReview forum: "Towards trustworthy predictions from deep neural networks with fast adversarial calibration"
_ICLR.cc/2020/Conference — Reject_

### Official Review · AnonReviewer1 · 2019-10-22
**Official Blind Review #1**

**Rating:** 3

**Review:**

This paper proposed FALCON, a simple method to produce well-calibrated uncertainty estimation. The idea is to introduce two additional terms, one that directly encourage lower confidence for all negative classes of all data points, and another one that optimizes the ECE for adversarial samples. Experiments show that FALCON outperforms several state-of-the-art methods for calibrating neural network predictions.

Although the first term, L_S, affects only negative predictions, it is still somewhat strange to uniformly operate on all data points. It would help to perform an ablation study to see how L_S affect the results.

Some description in Section 2 could use more mathematical rigor (e.g., when describing L_{adv}).

The authors use EDL as a baseline, I was wondering why not use the more commonly used, and possibly more effective, temperature scaling (TS) method from Guo et al. 2017. Note that the EDL paper does not seem to explicitly compare EDL and TS.

Figure 4 (middle and right) is confusing. The line style is not consistent in the figures.

There are a few places where the text is rather vague and confusing. For example, what do you mean by ‘non-misleading evidence’ when describing L_{adv}? It would also be better to provide more insight more L_S to help the readers out. For example, it would help to state that L_S operate only on negative predictions.

Most baselines are rather simple non-probabilistic (non-Bayesian) methods. Besides, MNF, It would also be interesting to see how FALCON compare other probabilistic NN method such as natural parameter networks, where they also explicitly evaluated uncertainty estimation.


**Experience Assessment:**

I have published one or two papers in this area.

**Review Assessment: Checking Correctness Of Derivations And Theory:**

I assessed the sensibility of the derivations and theory.

**Review Assessment: Checking Correctness Of Experiments:**

I carefully checked the experiments.

**Review Assessment: Thoroughness In Paper Reading:**

I read the paper thoroughly.

---

> ### Author Response · Authors · 2019-11-13
> **We have added the ablation study and new baselines suggested by the reviewer and improved clarity.**
>
> We thank the reviewer for constructive suggestions and have revised the manuscript accordingly.
>
> Clarifications and consistency. We have followed the reviewer and corrected the linestyle in figure 4 and clarified explanations around L_S and L_{adv}.
>
> Additional experiments. We now present an ablation study, illustrating that while the individual terms yield substantial improvements over the L2-dropout baseline, the combination of both loss terms results in a consistently better ECE compared to one term only. In addition, we include two more baselines, namely temperature scaling and an additional Bayesian neural network (SVI based on flipout), as suggested by the reviewer. In concurrent work ([1] referenced by reviewer #3), Ovadia et al. show that temperature scaling, while improving in-domain calibration, is not beneficial for calibration under domain-drift, which is also confirmed by our new experiments. In addition, we observe that SVI results in a decreased accuracy compared to other baselines and while calibration under domain drift improves in comparison to L2-dropout, it is substantially worse than FALCON and also EDL (Table S1 and Figure S3 appendix A.2). In total we now evalute 4 probabilistic baselines: Dropout as Bayesian approximation, Multiplicative Normalizing Flows, Evidential Deep Learning (predictions are modelled by placing a Dirichlet distribution on the class probabilities) and SVI based on flipout.

---

### Official Review · AnonReviewer3 · 2019-10-23
**Official Blind Review #3**

**Rating:** 3

**Review:**

The paper presents a method for calibrating neural networks on in- and out-of-distribution data using two additional loss terms: entropy-encouraging loss term which maximizes softmax probabilities for wrong classes and adversarial calibration loss term which pushes confidences to match accuracy on adversarial examples. The idea of using adversarial calibration training is interesting and promising, however, the clarity of the paper needs significant improvement and there are several issues which need to be addressed; for this reason, I recommend a weak reject for the current version.

1. One of the contributions listed in the paper is that authors “illustrate the limitations of entropy as measure for trustworthy predictions and introduce a new metric to quantify technical trustworthiness based on the concept of calibration”; the section 2.1 discusses this in detail. The proposed metric is expected calibration error (ECE) averaged over different levels of noise perturbations applied to data during test time. This metric assumes that for a given dataset we know in advance what kind of noise perturbations can be implied to cause out-of-domain/domain-shift scenarios during test time. This assumption most likely doesn’t hold in real-world applications since domain shifts may be of various kinds. Moreover, calibration may not make sense at all for out-of-domain data if test inputs don’t belong to any of the train classes (as opposed to entropy of predictive distribution which still can be computed and is expected to be higher for such inputs). The metric is also dependent on the considered noise level range; the plot of ECE vs noise level can be illustrative and informative while the averaged value of ECE over all noise levels can be misleading (at least, we may want to have some discount factor to account for the fact that with noise level 0 we strongly care about calibration, while for very high noise levels calibration doesn’t give us much information since the useful patterns in data may be corrupted and it may be impossible even for humans to classify such objects, e.g. on Figure 1-top-left for noise level 100, the digit 6 is corrupted so much that we wouldn’t care about calibration for such inputs but just want to maximize the entropy).
The paper also claims “Recent efforts in terms of evaluating predictive uncertainty have focused on entropy as measure for uncertainty-awareness for predictions under domain shift.” In previous work addressing uncertainty in domain shift [1], not only entropy but other various metrics are considered for out-of-domain detection (Brier score, thresholded confidences, etc), these metrics don’t depend on particular perturbations and are very informative.
The introduced metric, ECE for different noise labels, makes sense only in the toy scenarios where we can control the noise level, however, it would still be better to either look at the plots or to use some discounted averaged ECE over different noise levels, but not equal average. So the significance of this contribution (introduction of new metric) is limited.

2. The clarity of the paper could be significantly improved.

(a) Figure 1: the top left plot suggests that for high noise level “wrong predictions are often made with high confidence” (so the entropy of this distribution is low) while the top right plot shows that on average predictive entropy gradually grows and is high for high noise levels, so the top right plot is probably not a representative example and it might be misleading to claim this is an “often” case.

(b) Could you please clarify what you mean by “after removing non-misleading evidence” in section 2.2.1? In the next sentence, the remaining probability is probably distribution uniformly across C-1, not C, classes. The predictive entropy loss term essentially maximizes the probabilities of wrong classes with a lower coefficient. How would you support the claim that the loss surface is unchanged?
The loss term is “parameter-free” but still we need to tune a coefficient lambda_S for it.

(c) The adversarial loss equation in section 2.2.2 is written as L2-norm of a scalar value, why is L2-norm needed? The acc(B_m) is not differentiable so it is probably just considered as constant in the paper? Please comment on that.

(d) Section 3.2 and Figure 4. On Figure 4 Middle and Right plots have different colors than those listed on legend, please, fix this. On the middle plot, both FALCON and EDL have high ECE for noise levels <50 which indicates that they are both highly underconfident for in-distribution data and low noise levels, while other methods have close to 0 ECE on low noise levels. The authors only comment on EDL underconfidence: “it is worth noting that EDL has a substantially higher ECE for in-domain predictions, reflecting under-confident predictions on the test set”, and not on FALCON underconfidence. However, using the proposed score, ECE averaged over all noise levels (Table 2), it may look like the method is doing a good job, while underconfidence problem is revealed when looking at Figure 4. This is also an illustration of the concern about the proposed metric I raised in point 1.

3.  It would help to have an ablation study in the main text of the paper showing the significance of each loss term: in the appendix Figure S1, it is shown that adding adversarial loss to standard and entropy loss helps. Is the entropy loss needed at all? How does performance change if we only have standard and adversarial loss? How sensitive is performance to the choice of lambda_avd and lambda_s?


[1] Ovadia, Yaniv, et al. "Can You Trust Your Model's Uncertainty? Evaluating Predictive Uncertainty Under Dataset Shift." arXiv preprint arXiv:1906.02530 (2019).

**Experience Assessment:**

I have published one or two papers in this area.

**Review Assessment: Checking Correctness Of Derivations And Theory:**

N/A

**Review Assessment: Checking Correctness Of Experiments:**

I carefully checked the experiments.

**Review Assessment: Thoroughness In Paper Reading:**

I read the paper at least twice and used my best judgement in assessing the paper.

---

> ### Author Response · Authors · 2019-11-13
> **Point-by-point response to concerns regarding ECE.**
>
> We thank the reviewer for providing constructive comments and have attempted to address all issues in full. Below we provide a point-by-point response to all raised concerns. We are more than happy to provide further clarifications, should new questions arise from our response.
>
> Unknown domain drift and noise levels. We agree that in real-world applications the type of domain drift is not known/can be of various kinds - this is why we evaluate FALCON as well as various baselines on diverse types of domain drifts that were not seen during training, including different image perturbations but also character swaps in an NLP task; we argue that FALCON outperforming other methods on this diverse set of domain drifts is evidence that it is likely to also have a superior performance for other unknown types of domain drift. Since for practical applications neither the type of domain drift nor the amount of domain drift (noise level) is known in advance we also do not introduce a weighting of individual noise levels, but decided to use an unweighted average across all noise levels.
>
> Calibration for high levels of noise and truly OOD data. For very high levels of noise and in particular inputs that do not belong to any of the train classes, entropy can indeed be a complementary relevant metric and we now present additional evaluations demonstrating that FALCON also outperforms state-of-the-art methods in terms of entropy for high noise levels (Fig. S7 in Appendix A.7 shows FALCON outperforms baselines on MNIST, CIFAR and sequential MNIST; Fig. S8 shows the distribution of confidece scores in the OOD/high noise scenario and illustrates that FALCON has lower confidence scores than all baselines in the OOD case). However, we disagree that we wouldn't care about calibration for high noise levels: for very high noise levels where it is impossible to make a correct prediction, the accuracy is (close to) random and in this case perfect calibration means maximum entropy (and is therefore still meaningful and desirable). So we argue that it is actually because we care about calibration that we want to maximize entropy for high noise levels (at least in domain-drift scenarios where we assume that the distribution the inputs are drawn from changes, but the set of labels remains unchanged). The advantage of motivating maximum entropy via calibration is that we can use the same framework for the entire continuous spectrum of domain drift without the need of introducing an arbitrary threshold from which on we treat samples as OOD and maximize entropy.  We now clarify that ECE is designed for such domain drift scenarios and discuss that in case of OOD samples with a different set of labels, entropy is a better metric, as suggested by the reviewer.
>
> Relation to ref [1]: Ovadia et al. evaluate - in an independent and concurrent submission - calibration under domain drift for a range of existing algorithms and datasets based on ECE, Brier score, and other metrics. The authors independently make the same argument as we do and argue that since the type of domain shift that will occur after training is unknown in practice, calibration should be quantified based on a range of different types of domain drift where the noise model can be controlled. They therefore generate shifted data for a range of perturbations and noise levels (16 distortions such as image blur at 5 noise levels) and compute ECE and Brier score across all noise levels and perturbations (section 4.2 in Ovadia et al.). It is worth noting that their findings confirm our results for baseline methods, in particular for deep ensembles and MC-dropout (they did not consider EDL and have not presented any new approach).

---

> ### Author Response · Authors · 2019-11-13
> **Pont-by-point response to concerns regarding additional experiments  (point 3)**
>
>  We have now extended our ablation study and report results on the relevance of both loss terms, demonstrating that the combination of both terms yields best results. In addition, we show that our method is robust to the choice of lambda_adv and lambda_s when varied over one order of magnitude (variation of micro-averaged ECE less than 0.04 when varying lambda_adv between 0.0005 and 0.03 and less than 0.06 when varying lambda_S between 10 and 100; Appendix A.4 - due to space constraints we could not move this to the main text).

---

> ### Author Response · Authors · 2019-11-13
> **Point-by-point response to concerns regarding clarity (point 2)**
>
> We thank the reviewer for suggestions on how clarity can be improved and have revised the manuscript accordingly.
>
> (a) "high confidence" refers to miscalibration in the sense that at high noise levels where accuracy approaches random levels (right panel), entropy should approach maximum entropy for calibrated predictions. In contrast, averaged over all predictions, entropy actually saturates at 0.5 (22% maximum entropy), indicating that wrong predictions are indeed often made with high confidence.  EDIT: New Figure S8 in the appendix shows that the median confidence score is 75.8% at the highest noise level, also confirming this frequent overconfidence.
>
> (b) We now clarify that for wrong predictions our loss encourages a uniform distribution across all C classes and weaken our claim regarding the loss surface, putting it in context with EDL (reconstruction loss remains cross entropy for FALCON and entropy loss and adversarial loss do not directly affect accuracy).
>
> (c) We thank the reviewer for pointing us to the L2  norm in in the description of the adversarial loss. Unfortunately in the initial submission we had a typo in the formula that is now corrected: in the actual implementation the L2  norm is taken of the difference between confidence and acc(B_m) across all samples in the minibatch (rather than of the scalar – we apologise for this mistake).  This was chosen to penalise greater deviations of the confidence from accuracy. The reviewer is also correct regarding acc(B_m): accuracy is computed before each update and treated as constant. We have now expanded the corresponding section and explain this in more detail; we have also reformulated the loss function to correct the L2  issue and yield a more easily interpretable formulation. Finally, we have extended Algorithm 1 to better reflect the computation of all loss terms (Appendix A.3).
>
> (d) We have fixed the legend. FALCON is indeed also slightly underconfident for small perturbations and we now discuss and illustrate this in more detail. In Figures S4 and S5 in Appedix A.6 we now directly compare the confidence levels for EDL and FALCON for y-zoom, both via the sample used in Fig. 4 and at the dataset level via empirical CDFs (more commonly used version of confidence-threshold plots). We illustrate that the slight under-confidence of FALCON has little practical relevance, whereas the high levels of overconfidence (Fig. 1, 2, 4) displayed by all other methods can be highly problematic and misleading in practice: an underconfident prediction of 89% rather than well-calibrated 95% is unlikely to affect a decision making process, whereas a strong overconfidence of 80% rather than well-calibrated 10% can be detrimental in many real world application (Appendix A.6, Fig. S4, S5, A.7 Fig. S8). We report that FALCON is consistently less underconfident than EDL in terms of ECE and further illustrate this via empirical CDFs of confidence scores: While for FALCON 13.3% of predictions on the test set are made with a  confidence of less  than 90%, this increases to 30.5% for EDL, which is substantially higher than the 7.7% we observe for the well-calibrated  Deep Ensembles (Fig. S4). In other words, only a minority of 5.6% of all predictions made by FALCON are made with a confidence of less than 90% and should have been made with a higher confidence.  EDIT: New Figure S8 in the appendix illustrating the distribution of confidence scores also shows that EDL is substantially more underconfident than FALCON with a mean confidence score of 83.4% for EDL and 94.9% for FALCON.

---

### Official Review · AnonReviewer2 · 2019-10-23
**Official Blind Review #2**

**Rating:** 3

**Review:**

This paper introduces a new loss function for training deep neural networks, which show good performance with respect to well-calibrated, trustworthy probabilities for samples after a domain shift. The authors conduct experiments with multiple datasets and multiple forms of perturbations, where the proposed method achieve superior performance.

I have the following major concerns with the paper:

1. Presentation: The paper's presentation is very weak. It contains repetitive, long, convoluted statements and paragraphs all throughout making it difficult for the reader to understand anything. The first time I read this paper, I couldn't process what is happening. The introduction is more like realted work with less focus on what they are trying to pitch in the paper. However, the latter is the less severe concern.

2. While I appreciate the use of reliability diagrams and the loss function inspired from it, I do not completely understand what are the entities in equations 1 and 2 or how they are used later. Please give them a name, and elaborate on them.

3. While I was reading the paragraph before section 2.2 the first time, it seems to me that ECE is defined for those 10 pertubations specifically. I believe this is not the case; hence, the authors should make it more general and divert the specific details to experiments.

4. I am not sure what the authors mean by, "... while the loss surface remains largely unchanged", in paragraph above section 2.2.2. I believe the authors have constructed a new loss function by adding a new term to it. That makes it difficult to understand the advantage the authors are talking about after this statement. Overall, I am not really convinced how in contrast to Bayesian Deep Learning, their approach can be EASILY applied to LSTM's and GRU's.

5. What is L_{adv}? There is no equation number, no discussion around where this is defined. Again a presentation issue.

Minor: Please clarify the reference for supplementary materials. For example, it is not clear that Table S1 is in supplementary material.

Overall, this paper is good and has an interesting idea. The experiments are also extensive useful. However, I have reservations regarding the presentation of this paper at this moment.

--- After Rebuttal  ---

I thank the authors for providing a detailed response to my questions and editing the paper. I am now more positive about the paper; however, I still feel the presentation of the paper could be further improved. At this point, I will not move to acceptance range and keep the same score.

**Experience Assessment:**

I have read many papers in this area.

**Review Assessment: Checking Correctness Of Derivations And Theory:**

N/A

**Review Assessment: Checking Correctness Of Experiments:**

I assessed the sensibility of the experiments.

**Review Assessment: Thoroughness In Paper Reading:**

I read the paper thoroughly.

---

> ### Author Response · Authors · 2019-11-13
> **We have now improved the presentation and revised the manuscript following the suggestions of the reviewer**
>
> We thank the reviewer for the constructive suggestions on how to improve the presentation and are glad to read the reviewer found our submission was "good" and "has an interesting idea". We have now improved the presentation and revised the manuscript following the suggestions of the reviewer.
>
> 1. In the introduction we now clarify where we describe related work and focus more on the "pitch".
>
> 2. We have now clarified and elaborated on all entities in both equations. Eq. 1 is the formal definition of calibration, leading to the definition of miscalibration/calibration error (eq 2). The expectation in eq 2 is then approximated as the expected calibration error (ECE) eq 3, which in turn is used throughout the manuscript to quantify calibration under domain drift and, as pointed out by the reviewer, motivates the FALCON loss term L_{adv}.
>
> 3. The reviewer is correct. ECE is defined independent of any type of specific domain shift and was indeed recently popularised when used to quantify in-domain calibration. Since in practical applications it is a priori not clear what type or magnitude of domain drift will occur, we evaluate calibration under domain drift for our approach (and numerous baselines) on 10 different perturbations and 10 different noise levels not seen during training that were chosen to represent different ways of domain drift. We have now clarified this and moved details to Experiments as suggested.
>
> 4. We agree that our statement regarding the loss surface was not very clear and worded too strongly. We have rephrased this section to reflect that in comparison to probabilistic methods such as EDL or MNF, the reconstruction loss remains unchanged from the standard categorical cross entropy loss. Our additional loss terms do not directly affect accuracy and we hypothesize that this may be a reason why in contrast to EDL our approach yields SOTA accuracies also for more difficult learning problems (e.g. sequential MNIST with LSTMs, Tables S2 and S3 in Appendix A.7). We also agree that application to LSTMs/GRUs/more complex architectures depends on the type of BNN – while MC-dropout is straight-forward to apply, more complex state-of-the-art approaches such as multiplicative  normalizing flows are very hard to apply to architectures that contain layers that are more complex than dense layers or convolutional layers. For example for LSTM cells but also ResNET architectures, first, forward passes using local reparameterizations to yield a posterior based on multiplicative normalising flows have to be derived from scratch in a non-trivial manner and second, result in a high computational complexity. In contrast, FALCON only requires  the addition two simple  loss terms to the existing cross entropy loss. We now clarify this in the manuscript.
>
> 5. L_{adv} is the adversarial loss term which we now motivate and describe more extensively in section 2.2.2; for improved clarity we have now also expanded the formula along with a more detailed discussion around the definition.

---

### Author Response · Authors · 2019-11-13
**Response to all reviewers**

We thank all reviewers for their constructive comments and we have addressed individual concerns in point-by-point responses below.
Following the suggestions of all reviewers we have revised the manuscript to improve clarity and mathematical rigor, in particular in section 2 where we introduce ECE and our loss terms.
In addition, we have also added the additional experiments suggested by the reviewers: We have now extended our ablation study to analyse the influence of both loss terms and the robustness of FALCON wrt hyperparameter choice. These experiments illustrate that while individual loss terms yield improvements over many baselines, the combination results in consistently better calibration under domain drift.
Furthermore, we have added experiments on two additional baselines, namely temperature scaling and SVI and show that FALCON also yields a better calibration under domain drift also compared to these modelling  approaches.
Finally, we have added further evaluations and first demonstrated a better OOD perfromance of our method over baselines using the entropy and boxplots illustrating overall distribution of confidence scores.  Second, we further assessed limitations of our method related to under-confident scores using empirical CDFs.

We hope we could address all concerns in a satisfactory manner and hope that the reviewers will re-evaluate their scores in light of our response and revision.

Our revisions have resulted in the following changes in the manuscript:
Section 1: minor changes to make separation in related work and our contribution explicit (following Reviewer2).

Section 2.1.1:
- Add mathematical rigor and elaborate on eq. 1 and 2 (R2)
- Move description of specific perturbations to Experiments (R2)
- Add comment on limitations of ECE for high noise levels and change in label set (R3)

Section 2.2.1:
- Replace the formulation on misleading evidence and clarify that loss term only operates on wrong/negative predictions (R3, R1)
- Remove our claim that the  “loss surface remains largely unchanged” and instead clarify that the loss terms do not directly affect the reconstruction loss and speculate that this is a reason why in contrast to e.g. EDL our model generalises to more complex learning problems/architectures (R2, R3)

Section 2.2.2:
- Add mathematical rigor and more details when introducing loss term L_adv (R1, R2, R3)
- Add reference to ablation studies and robustness analysis (R3,R1)

Section 3: Add reference to additional baselines

Section 3.1: Move former Table 1 to Appendix (now table S2) to make space for clarifications suggested by the reviewers.

Section 3.2:
- Add discussion on underconfidence of EDL and FALCON for small perturbations and reference appendix A.6 (R3)
- Add discussion on OOD performance and quantification via entropy (R3)

Section 4: Shorten the discussion to make space for the clarifications suggested by the reviewers.

Appendix:
- Algorithm 1: add new states to clarify computation of L_adv
- A.4: Extend ablation study to second loss terms and add robustness analysis (R3, R2)
- A.5: Add results for additional baselines
- A.6: Add discussion on under-confidence of EDL and FALCON for small perturbations; add empirical CDF and boxplots of confidence scores for perturbation 0 (test set) to further quantify and illustrate extend of under-confidence and demonstrate it only affects a small fraction of samples for FALCON (R3)
- A.7 Add figure S7 showing entropy of all baselines under max. perturbation (OOD samples) for MNIST, CIFAR and sequential MNIST to demonstrate FALCON also outperforms baselines in terms of entropy (R3).  Add Figure S8 to further illustrate this with boxplots depicting the overall distribution of OOD confidence scores.

---

### Decision · Program_Chairs · 2019-12-19

**Decision:**

Reject

**Comment:**

This paper proposes an algorithm to produce well-calibrated uncertainty estimates. The work accomplishes this by introducing two loss terms: entropy-encouraging loss and an adversarial calibration loss to encourage predictive smoothness in response to adversarial input perturbations.

All reviewers recommended weak reject for this work with a major issue being the presentation of the work. Each reviewer provided specific examples of areas in which the paper text, figures, equations etc were unclear or missing details. Though the authors have put significant effort into responding to the specific reviewer mentions, the reviewers have determined that the manuscript would benefit from further revision for clarity.

Therefore, we do not recommend acceptance of this work at this time and instead encourage the authors to further iterate on the manuscript and consider resubmission to a future venue.